# Distribution and Genetic Diversity of *Aedes aegypti* Subspecies across the Sahelian Belt in Sudan

**DOI:** 10.3390/pathogens10010078

**Published:** 2021-01-17

**Authors:** Sara Abdelrahman Abuelmaali, Jamsari Amirul Firdaus Jamaluddin, Kheder Noaman, Mushal Allam, Hind Mohammad Abushama, Dia Eldin Elnaiem, Intan Haslina Ishak, Mustafa Fadzil Farid Wajidi, Zairi Jaal, Nur Faeza Abu Kassim

**Affiliations:** 1School of Biological Sciences, Universiti Sains Malaysia, Penang 11800, Malaysia; Sarabuelmaali@Student.usm.my (S.A.A.); jamsari_85@yahoo.com (J.A.F.J.); intanishak@usm.my (I.H.I.); 2National Public Health Laboratory, Federal Ministry of Health, Khartoum 11115, Sudan; 3National Center for Research, Tropical Medicine Research Institute, Khartoum 1304, Sudan; khedernoaman@gmail.com; 4National Institute for Communicable Diseases, National Health Laboratory Service, Johannesburg 2131, South Africa; mushalallam@gmail.com; 5Department of Zoology, Faculty of Science, University of Khartoum, Khartoum 321, Sudan; hindabushama@hotmail.com; 6Department of Natural Sciences, University of Maryland Eastern Shore, Maryland, MD 21853, USA; dialnaiem@gmail.com; 7Molecular Entomology Research Group, School of Distance Education, Universiti Sains Malaysia, Penang 11800, Malaysia; mfadzil@usm.my; 8Vector Control Research Unit, School of Biological Sciences, Universiti Sains Malaysia, Penang 11800, Malaysia; zairi@usm.my

**Keywords:** mitochondrial DNA, cytochrome oxidase-1 (CO1), *Aedes aegypti aegypti*, *Aedes aegypti formosus*, haplotype, Sudan

## Abstract

*Aedes aegypti* is the most important arboviral disease vector worldwide. In Africa, it exists as two morphologically distinct forms, often referred to as subspecies, *Aaa* and *Aaf*. There is a dearth of information on the distribution and genetic diversity of these two forms in Sudan and other African Sahelian region countries. This study aimed to explore the distribution and genetic diversity of *Aedes aegypti* subspecies using morphology and Cytochrome oxidase-1 mitochondrial marker in a large Sahelian zone in Sudan. An extensive cross-sectional survey of *Aedes aegypti* in Sudan was performed. Samples collected from eight locations were morphologically identified, subjected to DNA extraction, amplification, sequencing, and analyses. We classified four populations as *Aaa* and the other four as *Aaf*. Out of 140 sequence samples, forty-six distinct haplotypes were characterized. The haplotype and nucleotide diversity of the collected samples were 0.377–0.947 and 0.002–0.01, respectively. Isolation by distance was significantly evident (r = 0.586, *p* = 0.005). The SAMOVA test indicated that all *Aaf* populations are structured in one group, while the *Aaa* clustered into two groups. AMOVA showed 53.53% genetic differences within populations and 39.22% among groups. Phylogenetic relationships indicated two clusters in which the two subspecies were structured. Thus, the haplotype network consisted of three clusters.

## 1. Introduction

*Aedes aegypti* (*Ae. aegypti*), which is commonly known as the yellow fever mosquito, is recognized for the transmission of the most significant arboviral diseases, including dengue, chikungunya, and zika viruses [1,2] It is estimated that 70% (831 million) of the African population is vulnerable to arboviral disease infections [3].

This species is a tropical and subtropical mosquito with distribution throughout the globe but native to Sub-Saharan and African Sahelian regions, including Senegal, Cameroon, Kenya, Nigeria, Morocco, Western Sahara, Algeria, Tunisia, Egypt, and Sudan [4,5]. Known as the domestic mosquito, *Ae. aegypti* feeds on humans (anthropophilic) during the daytime hours. It also rests at indoor sites and breeds within and around the human environment, particularly in man-made containers (e.g., water jars, barrels, and tires) [6,7].

Unlike other continents, two forms or subspecies are known in Africa; *Ae. aegypti aegypti* (*Aaa*) and *Ae. aegypti formosus* (*Aaf*). *Ae. aegypti aegypti* is a light pale brown color form with white abdominal scales. It is known as the domestic subspecies, with strict breeding in human-made artificial containers and marked anthropophagy. On the other hand, *Ae. aegypti formosus* is darker in color and lacks the white abdominal scales. This form/subspecies is more sylvatic, breeding in natural habitats, such as tree holes, and more zoophilic [8,9]. The dark form (*Aaf*) is confined entirely to Africa, the south of the Sahara, its northern borders in Sudan, as shown by Lewis [10]. It has been reported from many parts of Africa, including Liberia, Kenya, Senegal, Ghana, Uganda, and Sudan. It has been also suggested that the dark form is perhaps abundant all over Africa [11,12,13,14,15].

The *Ae. aegypti* subspecies are known to vary in their disease transmission capacity, *Aaa* populations have a considerably higher vector competence for both dengue virus (DENV-2) and YF virus than *Aaf* [16,17]. Since the two subspecies show different susceptibilities to dengue viruses, it is important to understand their distribution and role in disease transmission. Therefore, a number of genetic markers have been developed to understand genetic variations in *Aedes aegypti* vectors and the genetic structure of its populations, and the phylogenetic and genetic diversity of *Aedes aegypti* have been characterized in different parts of the world since 1971 [18]. These markers include both biochemical and molecular tools, such as allozymes, nuclear DNA, microsatellites, and mitochondrial DNA [14,19,20,21,22,23]. Due to their maternal inheritance and rapid divergence, the mitochondrial DNA markers have been used extensively in phylogenetic, evolutionary, and population genetics studies of *Aedes aegypti* worldwide [24]. However, little has been done on the genetic structuring of the two subspecies in the African Sahelian region.

In Sudan, *Ae. aegypti* was described for the first time in Khartoum by Balfour (1903) [25]. In a subsequent study, the vector was reported widely in various geographical localities in Sudan including the eastern regions (Port Sudan and Kassala), central Sudan (Wad Medani and Khartoum), western regions (Al Fasher, and Al Junaynah), and the Nuba mountains in the south [8,25,26]. More recently, several studies documented the presence of *Aedes aegypti* in endemic foci of dengue, chikungunya, and yellow fever. The absence of other possible vectors in these locations suggests the definitive role of *Ae. aegypti* in the transmission of the viruses causing these diseases [27,28,29]. However, no data were published on infections of the viruses in the vector in the area.

Since the 18th century and for decades, Sudan has suffered from many arboviral diseases [30]. In 1955, Lewis emphasized the role of *Aedes aegypti* in yellow fever and dengue outbreaks in the country [10]. This notion was supported by several subsequent reports since the species was found in all areas where arbovirus outbreaks have occurred [27,30,31,32,33]. Dengue fever viruses have existed for decades, particularly in the eastern parts of the country [30,31,34]; however, recently, in 2015, the first dengue outbreak in the western part (Darfur) of the country occurred, with considerable other outbreaks having been mentioned in the country since 1908, and epidemics have continued to occur many times, causing extensive mortality [27]. Chikungunya viruses have also been reported in Sudan many times from different geographical areas [35,36,37,38] and a recent fatal chikungunya outbreak was reported in 2018–2019 in various regions in Sudan [30,39]. Yellow fever (YF) outbreaks have been reported in Sudan since 1940 and continue to occur, with severe outbreaks in recent decades [30,33]. 

Despite the crucial involvement of *Ae. aegypti* as a vector of yellow fever, dengue fever, chikungunya, and other arboviral diseases in Sudan, little is known about its distribution, population dynamics, genetic structure, and genetic variations across different endemic geographical areas of the country. Proper knowledge of these aspects should help to provide a better understanding of the epidemiology of arboviral diseases and their control [40]. 

This study was designed to investigate the distribution and genetic diversity of the two subspecies of *Aedes aegypti* across the Sahelian belt in Sudan. Using the cytochrome oxidase marker, we compared the genetic diversity within and between the two subspecies in different areas of Sudan. 

## 2. Results

### 2.1. Morphological and Molecular Identification of Ae. aegypti at Eight Sites of Sudan

We found *Ae. aegypti* in all eight study sites in Sudan. All samples were identified by morphological features, without ambiguities. Additionally, the species identification was positively confirmed by amplification and Rsal restriction enzyme analysis of the ITS1 region (700 bp) from ten *Aedes* mosquito samples. 

The morphological identification of mosquitoes collected in this study showed that both subspecies of *Ae. aegypti* are present in Sudan. The 100 adult females that emerged from samples taken from each site of four towns located in the eastern and central part of the county, namely Port Sudan (P), Tokar (T), Kassala (K), and Barakat/Gezira (G), were morphologically identified as the domestic form/subspecies *Aaa*, while the *Aedes* mosquitoes from the western and southern parts (Darfur and Kordofan) were identified as the wild form/subspecies *Aaf:* Nyala (N), Al Fasher (F), Al Junaynah (J) and Kadugli (D) (Table 1). The distribution of the two subspecies was differentiated between the western and the eastern parts of Sudan, with the White Nile and the main Nile separating the locations of their respective samples (Figure 1). 

The aquatic stages of *Aedes aegypti* were collected from different larval habitats inside homes or around human dwellings. The most dominant larval habitats that yielded most of the larvae for both *Ae. aegypti* subspecies were clay-pots, “Zeirr”, used for drinking water, followed by barrels and jerry cans, cement water reservoirs, tires, flower vases, old unused bathtubs, and other water containers (Table 1). 

### 2.2. Mitochondrial Haplotype Analysis

Following the amplification of an 860 bp fragment of the CO1 gene, the successfully sequenced samples were cleaned and trimmed to generate a final 603 bp fragment of CO1 sequences which was subjected to analyses. Forty-six distinct haplotypes were identified out of 140 female *Aedes aegypti* mosquitoes collected from all eight sites in Sudan. The numbers and diversity of these haplotypes are described in Table 2, and their relationships are illustrated in the median-joining haplotype network tree shown in Figure 2. Of these samples, nine haplotypes were shared by more than one individual mosquito, while 37 haplotypes were singletons, detected in a single mosquito (Table 2). 

There was a clear variation in the number of haplotypes in each study site (Table 2). In descending order, the numbers of haplotypes in different sites were 16 in Tokar, 14 in Kadugli, 8 in Nyala, 5 in Barakat, Kassala, and Port Sudan, 4 in Al Fasher, and 3 in Al Junaynah (Table 2). Each site had a number of unique haplotypes not found in other sites. In total, there were 33 site-specific haplotypes. On the other hand, two haplotypes (Hap 43 and 37) appeared to have a higher prevalence in all *Aaa* and *Aaf* populations, respectively (Figure 2). Hap 43 was found in 49.35% of all *Aaa* subspecies samples and occurred in all eastern Sudan study sites. In comparison, Hap 37 occurred in 27.14 % of *Aaf* samples (Figure 2) and was found in all western Sudan sites, except Al Junaynah. 

Three haplotypes were shared by some *Aaa* and *Aaf* populations. Hap 24 was found in two *Aaf* in Kadugli and two *Aaa* in Tokar. Hap 36 occurred in four *Aaa* samples in Barakat and four *Aaf* samples from Nyala, Al Fashir, and Kadugli. Hap 37 occurred in one *Aaa* sample in Tokar and 16 *Aaf* samples from Nyala, Al Fashir, and Kadugli (Figure 2).

Although the number of polymorphic sites and the nucleotide diversity were similar for the two *Ae. aegypti* subspecies (Number of polymorphic sites = 51 in *Aaa* and 52 in *Aaf*; Table 2), the level of haplotype diversity appeared to be consistently higher in *Aaf* (0.61 in Al Fashir, 0.7 in Al Junaynah, 0.92 in Kadugli and 0.933 in Nyala) than in *Aaa* (0.377 in Port Sudan, 0.539 in Kassala, 0.791 in Barakat and 0.947 in Tokar). Overall, Port Sudan appeared to have the lowest genetic diversity of *Aedes aegypti* compared to all other sites (Table 2).

### 2.3. Genetic Variance (Pairwise F_ST_) and Mantel Test Results

The overall subpopulation genetic variance (Pairwise F_ST_ values) showed values ranging from 0.015 (between Fasher and Kadugli *Aaf* populations) and 0.848 (between Fasher *Aaf* and Port Sudan *Aaa* populations) (Table 3). All between-population F_ST_ values were significant (*p* < 0.05), except between Kadugli and Barakat, Kadugli and Nyala, Kassala and Junaynah, Barakat and Junaynah, and Kadugli and Junaynah (Table 4).

The Mantel test revealed a medium to strong significant relationship between geographic distance and genetic differentiation correlation coefficient value (r) = 0.586, *p* = 0.005 (Figure 3). However, there were some deviations from this relationship, since some sites showed higher F_ST_ values with a nearer site than another site lying farther away. A case in point is the F_ST_ for the *Aaf* populations of Al Fashir and Kadugli (Distance = 553 km, and F_ST_ = 0.015) and Al Fashir and Junaynah towns (distance = 303 km, and F_ST_ = 0.366). Another example is the F_ST_ for Port Sudan and Tokar *Aaa* populations (Distance = 141 km; F_ST_ = 0.425) and in Port Sudan and Barakat populations (Distance = 700 km; F_ST_ = 0.389).

### 2.4. Genetic Structure of Aedes aegypti in Different Study Sites

The polar unrooted maximum-likelihood phylogenetic tree for 140 individuals *Aedes aegypti* from the eight study sites revealed two big clusters (Figure 4); most of the *Aaa* haplotypes clustered and appeared to be genetically structured, while the haplotypes of *Aaf* were represented by one group. It must be pointed out that there were some exceptions to this rule, as some samples from Tokar and Barakat/Gezira *Aaa* shared the same haplotypes with the *Aaf* populations haplotypes and clustered with them under the same cluster/branch (Figure 3).

Spatial analysis of molecular variance (SAMOVA) for population grouping also showed that the eight populations were grouped in three phylogeographically distinct groups/units. Group 1 includes Kadugli, Nyala, Al Fasher, and Al Junaynah *Aaf* populations. Group 2 consists of Port Sudan and Kassala *Aaa* populations. Group 3 consists of Barakat/Gezira and Tokar *Aaa* populations. Analysis of molecular variance (AMOVA) of the *Aedes aegypti* populations revealed 39.22% genetic variance among the three groups, and 53.53% within populations. (Table 5). 

### 2.5. Test of Neutrality and Natural Selection in Different Populations of Ae. aegypti

We used Tajima’s D and Fu’s F tests to decipher deviations from neutrality owing to natural selection or population expansion in different populations of *Ae. aegypti* sampled in the study. The results of Tajima were found to be positive in the three sites of Tokar, Barakat/Gezira, and Al Fasher, indicating balancing natural selection or population sub-structuring; however, it was negative in Port Sudan, Kassala, Kadugli, Nyala, and Al Junaynah, indicating a recent directional selection or recent population growth. The results of the Fu’s FS test were positive in Kassala, Barakat, Al Fasher, and Al Junaynah, which provides evidence indicating a recent population bottleneck or over-dominant selection, and negative in Port Sudan, Tokar, Kadugli, and Nyala, indicating a recent population expansion or genetic hitchhiking (Table 2). 

## 3. Discussion

The risk of emerging and re-emerging arthropod-borne viral (arboviral) infections is growing rapidly around the globe, particularly in the African continent [3]. Arboviral diseases have become a major health issue in Sudan. Over the last two decades, outbreaks of yellow fever, dengue, and chikungunya caused high mortality and morbidity in different parts of the country, particularly Port Sudan and Kassala in the east and Darfur in the west [30,31,34,41]. *Ae. aegypti* has been claimed as the principal vector of arboviruses responsible for these diseases. In Sudan, *Aedes aegypti* is reported to be the principal vector of yellow fever and dengue fever in different parts of the country, including Darfur, Kordofan, Port Sudan, and Kassala [26,28,30,33], However, there is a serious deficiency in recently published reports on the distribution and transmission of arboviruses by *Aedes aegypti* in different parts of Sudan. To our knowledge, this is the first published report on the distribution and genetic variations of *Aedes aegypti* subspecies/forms in different localities in the Sahelian belt of Sudan. 

Despite the discovery that *Aedes aegypti* may exist as two morphologically distinct subspecies/forms, little is known about their relative contributions to the transmission of arboviral diseases in Africa [8]. Although reports showed that *Aaf* has less vector competence than *Aaa*, the vectorial capacity is a product of several attributes, including vector distribution, abundance, longevity, human-biting index, and tolerance of harsh environmental factors [15,16,17]. The current observations on the exclusive presence of *Aaf* and its apparent domestication and attachment to human dwellings in western Sudan indicate that *Aaf* might be the sole vector of arboviral diseases in this region. This notion supports the previous report by Lewis who stated that under conditions of urbanization, *ssp. formosus* can become very largely domesticated and can have an increased contribution to the transmission of human arboviruses [8,25]. Similarly, Futami observed that the two subspecies were sympatric in both artificial and natural containers, suggesting adaptation of *Aaf* to various habitats [42]. Interestingly, in west Africa, a group of researchers found that populations of *Aaf* can be competent transmitters of Flaviviruses [15]. Although knowledge of the vector competence in the two subspecies is a crucial component in arbovirus epidemiology and control, research on this aspect has been extremely limited. Further work is, therefore, needed to compare the survival, fecundity, and human-biting index and vector competence of the two subspecies. 

Our results show that both subspecies of *Ae. aegypti* are found in the Sahelian belt of Sudan. No previous publications showed similar results clearly, as the reported literature did not address the distinction between the two subspecies/forms of *Aedes aegypti* in the country. However, in the older literature, Mattingly reported that Lewis found a dark form in Erkawiet in eastern Sudan and stated that, in personal communication, Lewis informed him that this specimen was likely an *Ae. aegypti formosus* subspecies [8]. The lack of distinction between the two forms is also an apparent feature of the literature from neighboring Sahelian countries, as no recent records on the differentiation of *Ae. aegypti aegypti* and *Ae. aegypti formosus* were reported from Eritrea, Ethiopia, Chad, and Central Africa Republic [12,13,15,42].

We found an interesting difference in the distribution of the two subspecies of *Aedes aegypti* in Sudan; *Aedes aegypti aegypti*, found in four study sites east of the Nile river, i.e., eastern and central Sudan, and *Aedes aegypti formosus* in the other four sites in the west. Although the Nile and the White Nile appeared to show a clear demarcation of the location of the *Ae. aegypti aegypti* and *Ae. aegypti formosus*, it is unlikely that these river systems provide true geographical barriers for the two subspecies. A possible explanation for this distribution may be offered by the contrasting soil and different ecological parameters. Despite sharing similar latitudes and rainfall patterns, eastern and western Sudan have marked differences in soil that affect water retention, moisture, and maximum temperature and humidity [43,44]. In eastern Sudan, the soil is mainly chromic vertisols deposited by previous volcanic activities in the Ethiopian Plateau. This soil is highly hydrophobic, losing moisture quickly after the end of the rainy season and thus resulting in higher maximum temperature, lower humidity, and lower annual NDVI (Normalized Vegetation Index) values. In contrast, the sandy soil of western Sudan has higher retention of moisture, resulting in higher lower max temperature, higher humidity, and higher NDVI value [43,44]. 

In this study, we did not encounter the sympatric presence of the two subspecies/forms of *Aedes aegypti.* However, the possibility of the sympatric presence of the two subspecies cannot be ruled out, since our study was not exhaustive enough and was limited by several logistical barriers. In other locations in Africa, different authors reported contrasting findings on the distribution of the two forms. Mattingly stated that the *Ae. a. formosus* was recorded in natural breeding sites in the forests or bush away from human dwellings in Kenya and Uganda [8]. On the other hand, Gloria-Soria mentioned two different distribution cases in Kenya. In one of these cases, populations of the two subspecies/forms were sympatric and freely mixed in Mombasa/Kenya. In the other case, in a geographically closely related site, they found that *Aaf* was restricted to the African forest, while *Aaa* was present in domestic habitats [14].

We found a relatively high genetic variation and structuring between different populations of *Aaa* and *Aaf* in Sudan. In the polar phylogenetic tree, two subspecies are clustered in two distinct groups with only three haplotypes shared between them. These results were further confirmed by SAMOVA analysis which separated the *Aaf* populations in one group while splitting the *Aaa* populations into two similar groups. Furthermore, the divergence between *Aaf* and *Aaa* populations was revealed in high F_ST_ values between the two subspecies, for example in the Port Sudan and Al Fasher sites (F_ST_ = 0.848).

Although all genetic variation indices of the mitochondrial gene cytochrome-oxidase 1 (CO1) revealed low gene flow between the two subspecies in Sudan and high genetic diversity between their populations, it is difficult to conclude whether this variation reflects a true difference between the two subspecies or the geographical distances that limited the gene flow. Other researchers, using microsatellite markers on sympatric and allopatric populations of *Aedes aegypti*, concluded that the two subspecies/forms are genetically distinct [14]. It may be hypothesized that possible gene flow between the two subspecies originated recently after the encroachment of the *Aaf* into the human habitat where the *Aaa* was already found. Using genetic data, Powell concluded that there was total isolation and lack of gene flow between the two subspecies about 400–550 years ago [45].

Our results contrast with the findings of Powell and Tabachnick [18], who reported that *Aaf* subspecies have less genetic variation than the *Aaa* subspecies. In our studies, we found that both subspecies have similarly high genetic variations. The difference between our results and the findings of [13] and [18] may be due to the ecological conditions. Our collection sites of *Aaf* populations were generally more arid than the sites of these authors. Coupled with the large geographic distances between our study sites, the high aridity limits the movement and gene flow between the populations.

It was evident that the level of genetic variation of *Aedes aegypti* populations was different in different sites. The lowest variation was observed in the *Aaa* population in the coastal city, Port Sudan. Similar results were reported by Elnour [29], who reported the presence of four haplotypes of mitochondrial CO1 gene in *Aedes aegypti* populations in Port Sudan. This lower genetic variation in Port Sudan may be due to the fact that the Red Sea provides consistent high humidity throughout the year, thus reducing the impact of dryness on the vector population. In contrast, other sites in the country had a higher impact of natural selection, as predicted by Tajima D and Fu’s Fs statistics. 

In conclusion, our results show that both subspecies of *Aedes aegypti* are found in Sudan, with clear variation in their distribution. Whereas the *Aaa* subspecies appears to be more abundant in the eastern part of the country, the *Aaf* subspecies is the main form of the vector found in western Sudan. There was a clear distinction in the genetic structure of the two subspecies. Further work is urgently required to elucidate the role of each subspecies in the transmission of dengue fever, chikungunya, yellow fever, and other arboviruses in Sudan and other countries of the African Sahelian region. Furthermore, work is needed to understand the ecological determinants of the distribution of the two subspecies and develop suitable control programs for the viruses they transmit. 

## 4. Methods

### 4.1. Study Sites

Mosquitoes were collected from eight study sites in Sudan, where dengue and other arboviral disease outbreaks were reported. As shown in Figure 1, the sites consisted of Port Sudan (Red Sea state), Kassala (Kassala state), Tokar (Red sea state), Barakat (Gezira state), Kadugli (South Kordofan state), Al Fashir (North Darfur state), Nyala (South Darfur state) and Al Junaynah (West Darfur state).

Seven of the study sites (Kassala, Tokar, Barakat, Kadugli, Nyala, Al Fashir, and Al Junaynah) have a tropical continental climate, characterized by a long (9 month) dry season between October and June and a short rainy season between July and September. The eighth site (Port Sudan), located on the coast of the Red Sea, has a hot desert climate with high levels of humidity and a short rainy season during the cooler months of November to February [43]. 

Apart from Barakat, all study sites have a consistent history of dengue fever, which results in more frequent outbreaks in Port Sudan, Tokar, and Kassala. Other major arboviral diseases in the study sites are yellow fever in Kadugli, Al Fashir, Nyala and Al Junaynah, and chikungunya, which was recently reported in all areas of the country. 

### 4.2. Mosquito Collection, Rearing, and Morphological Identification

Immature stages of *Aedes* mosquitoes (eggs, larvae, and pupae) were collected from their natural habitats indoors and outdoors in all the study sites throughout (January 2014–April 2017). Different breeding sites were surveyed. The samples were transferred to the insectary at the Sudan national public health laboratory for rearing at the optimum temperature (25 ± 2 °C) and relative humidity (80–90%) with a photoperiod of 12:12 (L:D) until the adults emerged. Samples from each collection were morphologically identified to their species using appropriate taxonomic keys [46]. After the adult’s emergence, *Aedes aegypti* females and males were classified into their subspecies/forms, either *Aaa* or *Aaf*, according to the appropriate morphological key, depending on the scale pattern system [47]. The mosquitoes with white scales on the first abdominal tergite were classified as *Aaa* form, while adults that possessed white scales in the first abdominal tergite were classified as *Aaf* form. Female mosquitoes of *Aaa* and *Aaf* from different sites were preserved individually in labelled microfuge tubes with 70% isopropanol and then kept in a −20 °C freezer. Preserved samples were transferred to the Universiti Sains Malaysia (USM) where the DNA extraction was performed.

### 4.3. DNA Extraction and Molecular Identification

Genomic DNA was extracted from single female mosquitoes using Qiagen DNeasy Blood and Tissue Extraction Kit (Qiagen, Hilden, Germany) following the manufacturer’s instructions with slight modifications in extending the incubation period to overnight at 65 °C. Extracted DNA was eluted in nuclease-free water and preserved in a −20 °C freezer until use. The DNA quantity and quality for all extracted samples were checked using Nanodrop Quawell UV spectrophotometer Q3000. Further confirmation of species identification was executed using internal transcribed spacer region 1 (ITS1) using forward primer ITS1A, 5 -CCT TTG TAC ACA CCG CCC GTC G-3, and reverse primer ITS1B, 5-ATG TGT CCT GCA GTT CAC A-3 described by [48] with modification in the cycling condition of 95 °C for 2 min followed by 40 cycles of 95 °C for 45 s, 53 °C for 40 s, 72 °C for 10 min and final extension of 72 °C for 5 min. Gel electrophoresis was performed on 1.2% agarose gel in Tris-acetate-EDTA buffer (TAE). PCR products containing DNA fragments were further digested by restriction analysis by taking 10 μL of the PCR product in a microfuge tube, 5 μL of 10× Rsal buffer, and 1 U/1 μL of Rsal enzyme per reaction (NEB II, New England Biolabs). After mixing, the tubes were incubated at 37 °C for 2 h and then separated on a 3.0% agarose gel stained with Red Safe TM Nucleic Acid Staining Solution (INTRON Biotechnology, Seongnam, Korea), and ran at 100 V for 30 min. The gel was viewed under UV light, and the photo was captured to determine the pattern of the fragment DNA segments following [48].

### 4.4. PCR Amplification for Mitochondrial Marker

Cytochrome oxidase 1 mitochondrial marker was used to detect polymorphisms among the different populations; a minimum of 10 mosquito samples from each site were tested (Table 1). Partial CO1 gene was amplified using primer pair (COI-F) 5’-TGTAATTGTAACAGCTCATGCA-3’ and (COI-R) 5’-AATGATCATAGAAGGGCT GGAC- 3’, mixed in 50 μL reaction volume containing 10 μL of 10× Green Buffer Go Taq (Promega, Madison, WI, USA), 3 μL of 25 mM MgCl_2_, 1 μL of 25 mM dNTP, 0.3 μM from each primer, 0.5 μL of Taq polymerase and 2 μL (>50 ng) from the sample DNA [49]. The thermal cycling conditions were configured as the first denaturation step at 94 °C for 2 min, followed by 35 cycles at 94 °C for 1 min, 54 °C for 30 s, and 72 °C for 1 min, and a final extension of 72 °C for 10 min. PCR products were separated on a 1.5% agarose gel electrophoresis in Tris–borate-EDTA buffer (TBE) for one hour, a 100 bp DNA ladder was used as a marker, and the agarose gel was stained with Red Safe dye. Then, the gel was viewed under UV light using the gel documentation system. PCR-amplified samples were purified from the agarose gel using a Qiagen purification kit following the manufacturer protocol [49]. 

The cleaned PCR products were chosen, isolated, and sent to Apical sequencing services (Kuala Lumpur, Malaysia) for the Sanger DNA sequencing process. The sequencing services were done with the ABI 3130XL automatic sequencer (Applied Biosystem, Foster City, CA, USA). 

### 4.5. Mitochondrial Sequences Analyses

Sequences were generated and checked manually by using Molecular Evolutionary Genetics Analysis software MEGA v7.0 [50] and compared to the previously published sequences in the Genbank database using the Basic Local Alignment Search Tool (BLAST, https://blast.ncbi.nlm.nih.gov/Blast.cgi). The nucleotides obtained were aligned by using the sequence alignment tool ClustalW in the MEGA. Sequences were translated to amino acids (invertebrate mitochondrial code) to check for unexpected stop codons or frameshift mutations implemented in the same software. 

Unique haplotypes in the aligned matrix for both forms of *Aedes aegypti* were determined using DNAsp 5.10.1 [51] and were deposited in GenBank with the accession numbers MT193027 to MT193072.

### 4.6. Phylogenetic Analyses/Trees

The relationship between individuals of all population studies was illustrated using polar unrooted maximum-likelihood phylogenetic trees built using IQ-TREE 1.6.9 [52] and shaped using Figtree v1.4.0 [53]. The best model was found to be HKY+F+I using Model Finder [54]. Relationships between the different haplotypes in the eight sites from the two forms located in the different study sites were displayed using median-joining haplotype networks. Haplotype networks were drawn using NETWORK 10.00 [55]. 

### 4.7. Genetic Diversity and Genetic Structure of Aedes aegypti Subspecies

Standard molecular diversity indices—the number of haplotypes per locality (h), the number of variable sites (S), haplotype diversity (Hd), and nucleotide diversity (*π*)—were estimated within *Aaa* and *Aaf* populations using Arlequin v3.5 [56].

Genetic diversity and molecular variance of the populations assessed using Analyses of Molecular Variance (AMOVA) in Arlequin v3.5 [56]. The pairwise genetic distances (F_ST_) were estimated between the different populations and their significance.

Geographical distances between the different study sites were approximated using a ruler tool in Google Earth TM (Google Inc., Mountain View, CA, USA). Genetic isolation by distance (IBD) was assessed using pairwise geographical distance and was matched to pairwise F_ST_ in the different haplotypes independently to obtain a linear regression relationship. The Mantel test was carried out to prove the significance of the association between genetic variations and geographical distance between study sites using Arlequin v3.5 [56].

The genetic structure was estimated using Spatial Analyses of Molecular Variance (SAMOVA), SAMOVA v2.0 [57], to interpret the genetic barriers on population groups after the best-fit grouping pattern was estimated. The method classified the populations that were geographically similar and excellently separated the groups with the highest F_CT_ (diversity between groups) while the lowest F_SC_ (diversity among populations within a group) were considered the most possible populations grouping. Since the F_CT_ values in the study were found to have no significant difference, other methods were used following [58]. The F_CT_ (genetic differentiation among groups) was found to be relatively stable with very low differences; nevertheless, F_SC_ (genetic differentiation between individuals within such groups) revealed a noticeable decrease from two to three clusters of groups); from this, we conclude that the best grouping (best homogeneity among groups) is three-group clustering.

After that, SAMOVA structuring results were used to estimate the genetic molecular variance using Analyses of Molecular Variance (AMOVA) in Arlequin v3.5 [56].

Tajima’s D and Fu’s F statistics were used to detect deviation from neutrality and population expansion using Arlequin v3.5 [56].

## Figures and Tables

**Figure 1 pathogens-10-00078-f001:**
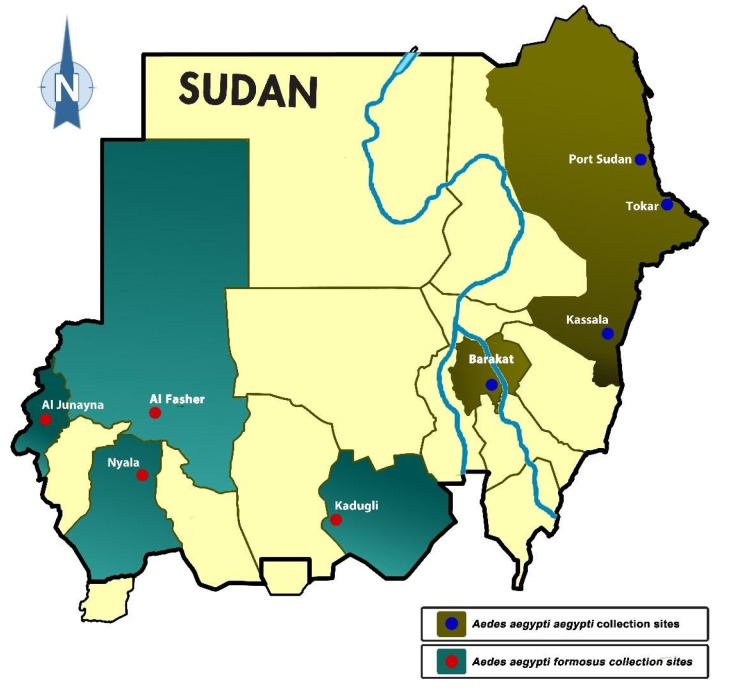
Map of Sudan showing the location of two subspecies/forms of *Aedes aegypti* collected in 8 sites. Note all *Ae. aegypti aegypti* were found in the east, whereas *Ae. aegypti formosus* was found in the west, with the main Nile and the White Nile separating their collection sites.

**Figure 2 pathogens-10-00078-f002:**
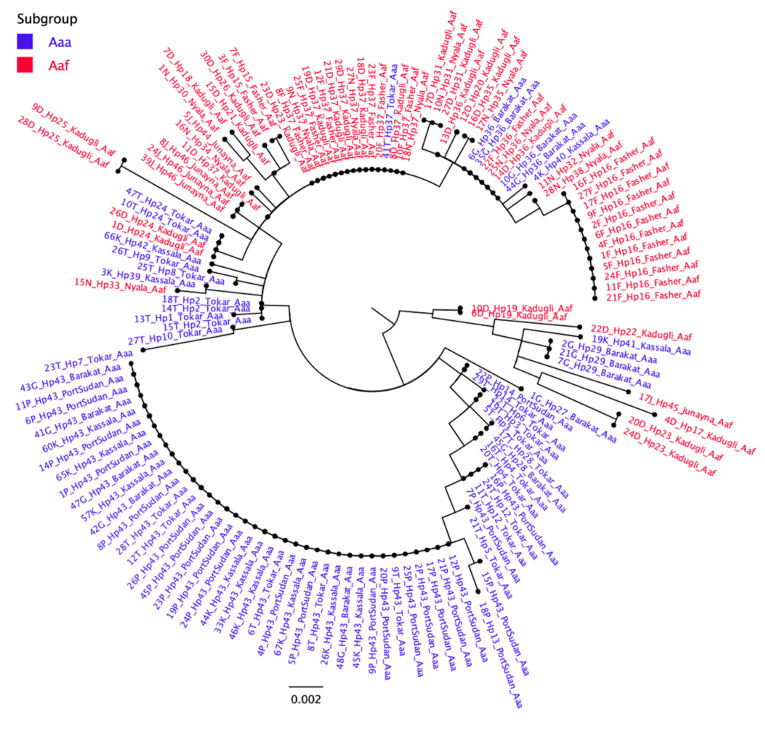
Median-joining haplotype network tree of *Ae. aegypti* subspecies/populations from 8 sites in Sudan. The tree was constructed using 46 haplotype CO1 mtDNA sequences from the eight study sites of Sudan. The size of each circle indicates the frequency of the incidence of each haplotype in the study populations. Between two haplotypes there is a minimum of one mutation. The network consists of two Groups including haplotypes from both *Aaa* and *Aaf* populations.

**Figure 3 pathogens-10-00078-f003:**
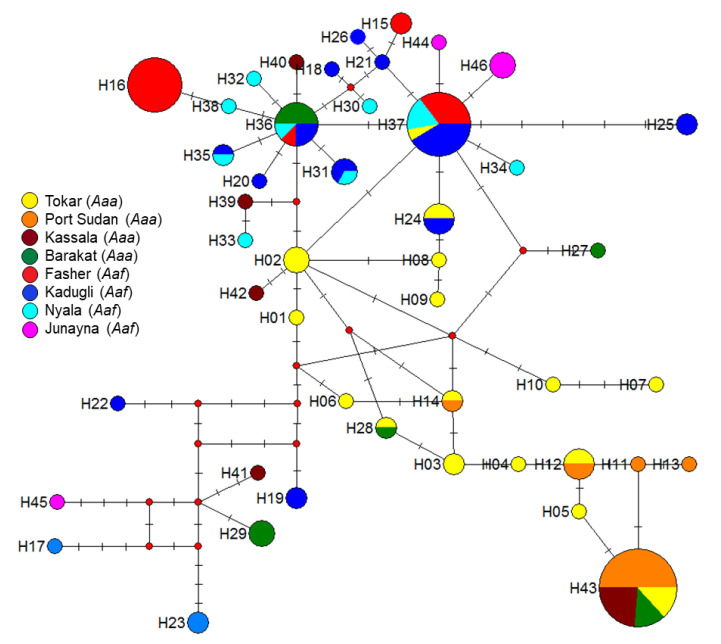
Polar maximum likelihood phylogenetic tree from 140 female *Aedes aegypti aegypti* (*Aaa*) and *Aedes aegypti formosus* (*Aaf*) form eight study sites in Sudan. Different colors indicate different subspecies/forms. Note most *Aaa* clustered in one group (blue colored), while *Aaf* clustered separately (red-colored). Exceptions to this were some *Aaa* from Tokar and Barakat clustered with *Aaf* populations.

**Figure 4 pathogens-10-00078-f004:**
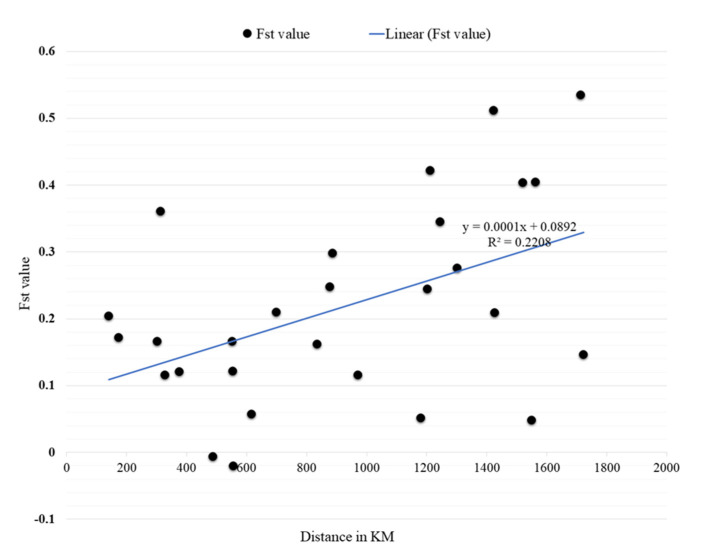
Results of the Mantel test on the correlation between genetic (F_st_) and geographical distances (KM) of *Aedes aegypti* samples collected in 8 locations in Sudan.

**Table 1 pathogens-10-00078-t001:** Samples collection sites, their numbers, and the subspecies distribution, site coordinates, and larval habitats.

Code	Site	Region	Subspecies	Coordinates	Collection Year	Larval Habitats	NO
**P**	Port Sudan	Coastal/Eastern	***Aaa***	19.617°37′0″ N, 37.217°13′0″ E	2014	Clay pots, cement water reservoir, plastic barrels and jerrycans	26
**T**	Tokar	Eastern	***Aaa***	18.425°25′31″N, 37.729°43′45″E	2016	Clay pots and jerrycans.	22
**K**	Kassala	Eastern	***Aaa***	15.45°27′0″N, 36.4°24′0″E	2014	Flowerpots, cement reservoir, and clay pots.	13
**G**	Barakat/Gezira	Central	***Aaa***	14.314°18′50.84″N, 33.534°32′3.74″E	2014	Plastic barrels, tyres, and bathtubs	20
**D**	Kadugli	South	***Aaf***	11.017°1′0″N, 29.717°43′0″E	2015	Tyres, clay pots, and plastic barrels	30
**N**	Nyala	West	***Aaf***	12.036°2′11″N, 24.878°52′37″E	2016–2017	Clay pots and jerrycans	16
**F**	Al Fasher	Northern West	***Aaf***	13.631°37′50″N, 25.35°21′0″E	2017	Clay pots and jerrycans	22
**J**	Al Junaynah	West	***Aaf***	13.45°27′0″N, 22.45°27′0″E	2014	Clay pots, cement barrels, and plastic containers	8

NO: number of mosquitoes used in population diversity study, *Aaa: Aedes aegypti aegypti, Aaf: Aedes aegypti formosus.*

**Table 2 pathogens-10-00078-t002:** Molecular diversity indices and neutrality tests of CO1 mitochondrial sequences of *Aedes aegypti* subspecies/forms from eight study sites in Sudan.

Site	Subspecies/Form	N	S	H	Hd	π	Tajima’s D	Fu’s F_S_
**Port Sudan**	*Aaa*/domestic	24	6	5	0.377	0.002	−1.319	−1.142
**Tokar**	*Aaa*/domestic	25	14	16	0.947	0.007	0.592	−6.792
**Kassala**	*Aaa*/domestic	13	17	5	0.539	0.008	−0.606	2.254
**Barakat/Gezira**	*Aaa*/domestic	14	14	5	0.791	0.010	1.500	3.419
**Kadugli**	*Aaf*/wild	27	26	14	0.920	0.009	−0.892	−2.899
**Nyala**	*Aaf*/wild	10	10	8	0.933	0.004	−1.507	−4.469
**Al Fasher**	*Aaf*/wild	21	5	4	0.610	0.003	1.076	1.690
**Al Junaynah**	*Aaf*/wild	5	11	3	0.700	0.007	−1.200	2.054

N, sample size; S number of polymorphic sites; H, number of haplotypes; Hd, haplotype diversity; π, nucleotide diversity; S, number of segregating sites; D, Tajima’s statistics; Fu’s F_S_ statistics.

**Table 3 pathogens-10-00078-t003:** Population divergence between samples (F_ST_ value) was performed in ARLEQUIN version 3.5 among eight populations of *Aedes aegypti* subspecies from Sudan.

Site (form)	P (*Aaa*)	T (*Aaa*)	K (*Aaa*)	G (*Aaa*)	D (*Aaf*)	N (*Aaf*)	F (*Aaf*)
**P (*Aaa*)**							
**T (*Aaa*)**	0.425						
**K (*Aaa*)**	0.125	0.158					
**G (*Aaa*)**	0.389	0.097	0.045				
**D (*Aaf*)**	0.652	0.248	0.396	0.186			
**F (*Aaf*)**	0.848	0.355	0.545	0.309	0.015		
**N (*Aaf*)**	0.837	0.421	0.603	0.384	0.160	0.191	
**J (*Aaf*)**	0.782	0.252	0.352	0.124	0.037	0.205	0.366

**Table 4 pathogens-10-00078-t004:** Probabilities (P values) of the Pairwise divergence (F_ST_) between eight populations subspecies of *Aedes aegypti* in Sudan.

Site (form)	P (*Aaa*)	T (*Aaa*)	K (*Aaa*)	G (*Aaa*)	D (*Aaf*)	N (*Aaf*)	F (*Aaf*)	J (*Aaf*)
**P (*Aaa*)**								
**T (*Aaa*)**	0.000							
**K (*Aaa*)**	0.021	0.014						
**G (*Aaa*)**	0.000	0.030	**0.172**					
**D (*Aaf*)**	0.000	0.000	0.000	0.001				
**N (*Aaf*)**	0.000	0.000	0.000	0.002	**0.267**			
**F (*Aaf*)**	0.000	0.000	0.000	0.000	0.000	0.014		
**J (*Aaf*)**	0.000	0.008	0.007	**0.087**	**0.156**	0.004	0.000	

Bold numbers indicate the nonsignificant at *p* > 0.05.

**Table 5 pathogens-10-00078-t005:** Analysis of molecular variance (AMOVA) of three groups of *Aedes aegypti* collected from 8 sites in Sudan.

Source of Variation	df	Sum of Squares	Variance Components	Percentage of Variation
Among Groups	2	131.012	1.310	39.22
Among Individuals within Groups	5	27.672	0.243	7.26
Within Populations	131	234.244	1.788	53.53
Total	138	392.928	3.341	

df = degree of freedom.

## Data Availability

Data is contained within the article.

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
