# Peer review of "Distribution and Genetic Diversity of Aedes aegypti Subspecies across the Sahelian Belt in Sudan"

_pathogens, 2021, doi:10.3390/pathogens10010078_

Round 1

Reviewer 1 Report

Comments for the MS entitled: Distribution and Genetic Diversity of Aedes aegypti subspecies across the Sahelian Belt in Sudan by Abuelmaali et al.

The authors investigate the distribution and genetic diversity of Aedes aegypti subspecies using morphology and cytochrome oxidase-1 mitochondrial marker in Sudan. The authors found a relatively high genetic variation and structuring between different populations of Aaa and Aaf in Sudan, two subspecies of Aedes aegypti in Africa.

Comment:

  1. It will make this study more promising if the authors could address the transmission of dengue virus or other Aedes borne viruses.
  2. Is it possible to address the difference on vector competence between these two subspecies?

Reviewer 2 Report

This is an interesting paper and it presents novel findings from an area of Africa previously neglected by researchers.  There are some things (minor and major) that need to be addressed, however.

Line 57: "The Ae. aegypti subspecies known to vary in their disease transmission capacity ...."  I believe this should read, "...subspecies are known...."

Line 102:  female adults, or adult females, not adults female.

Lines 110, 111, and elsewhere in the manuscript: "breeding sites" is not as good a term as "larval habitats".

Line 170:  I can find no reporting of results of Li tests, although they are mentioned here.

Table 1.  site coordinates, not sites coordinates.

Table 2.  Previously Fu tests were mentioned as being reported in this table, but I see only Tajima tests.

Line 191:  I would replace "while" with "whereas."

Nice discussion of the possible reasons for geographic separation of the two forms.

Line 291:  Replace 'arider" with "more arid"

Line 300:  were all of these tests reported, and discussed?

Line 326: delete "breeding" - natural habitats is enough.

Lines 414-415: The authors write, "After that SAMOVA structuring results will be used to estimate the genetic molecular variance analyses of the variance (AMOVA) in Arlequin v3.5...."  Why are the methods being described in the future tense?  This should be past tense.

Round 2

Reviewer 1 Report

The authors have addressed all my concerns.

Reviewer 2 Report

Two VERY small changes required.

Line 411: change to read "results were used"

Line 413: change to read "Fu's F"  instead of "Fu's Fs"

This are the only minor changes I see that need to be made.